# PandemonCAT: Monitoring the COVID-19 Pandemic in Catalonia, Spain

**DOI:** 10.3390/ijerph19084783

**Published:** 2022-04-14

**Authors:** Somnath Chaudhuri, Gerard Giménez-Adsuar, Marc Saez, Maria A. Barceló

**Affiliations:** 1Research Group on Statistics, Econometrics and Health (GRECS), University of Girona, 17004 Girona, Spain; chaudhuri.somnath@udg.edu (S.C.); ggadsuar@gmail.com (G.G.-A.); marc.saez@udg.edu (M.S.); 2CIBER of Epidemiology and Public Health (CIBERESP), 17003 Madrid, Spain

**Keywords:** COVID-19, mobility, spatiotemporal, shiny

## Abstract

Background: The principal objective of this paper is to introduce an online interactive application that helps in real-time monitoring of the COVID-19 pandemic in Catalonia, Spain (PandemonCAT). Methods: This application is designed as a collection of user-friendly dashboards using open-source R software supported by the Shiny package. Results: PandemonCAT reports accumulated weekly updates of COVID-19 dynamics in a geospatial interactive platform for individual basic health areas (ABSs) of Catalonia. It also shows on a georeferenced map the evolution of vaccination campaigns representing the share of population with either one or two shots of the vaccine, for populations of different age groups. In addition, the application reports information about environmental and socioeconomic variables and also provides an interactive interface to visualize monthly public mobility before, during, and after the lockdown phases. Finally, we report the smoothed standardized COVID-19 infected cases and mortality rates on maps of basic health areas ABSs and regions of Catalonia. These smoothed rates allow the user to explore geographic patterns in incidence and mortality rates. The visualization of the variables that could have some influence on the spatiotemporal dynamics of the pandemic is demonstrated. Conclusions: We believe the addition of these new dimensions, which is the key innovation of our project, will improve the current understanding of the spread and the impact of COVID-19 in the community. This application can be used as an open tool for consultation by the public of Catalonia and Spain in general. It could also have implications in facilitating the visualization of public health data, allowing timely interpretation due to the unpredictable nature of the pandemic.

## 1. Introduction

Catalonia, the second most populous autonomous community in Spain with 7.6 million inhabitants (Figure 1), as of 10 January 2022, has been the first most affected by the COVID-19 pandemic (the Madrid region being the second most affected), by number of cases (1,333,517 cases, 18.61% of all cases in Spain, 17,625 cases per 100,000 inhabitants, compared to 15,132 cases per 100,000 in Spain), and the second by number of deaths (16,144 deaths, 17.95% of all deaths in Spain, 213 deaths per 100,000 inhabitants, compared to 190 deaths per 100,000 in Spain) [1,2]. The geographical distribution of the spread of the pandemic has not been spatially homogeneous in the Catalan territory and important differences at the small-area level have been observed.

Catalonia is basically an urban region. Sixty percent of the population resides in 23 cities with more than 50,000 inhabitants, and 52% in 14 cities with more than 100,000 inhabitants [3]. These include the second-largest city in Spain, Barcelona, and 36 adjacent municipalities making up the Barcelona Metropolitan Area.

The administrative aggregation levels in Catalonia are the following, from highest to lowest: autonomous community (Catalonia), 42 ‘comarcas’ (county-like regions, hereafter referred to as regions), and 947 municipalities. In addition, the health aggregation levels are the following, from highest to lowest: autonomous community, 9 health regions, 20 health sectors, and 343 health basic areas (ABS, for their acronym in the Catalan language).

According to the definition by Catalan health planning, an ABS is defined as the basic geographical unit through which primary health care services are coordinated in [4]. General practitioners, pediatricians, dentists, nurses and nursing assistants, social workers, and administrative support professionals make up each ABS’s Primary Care Team. In the 343 ABSs in Catalonia, with populations ranging from 371 to 72,321 people (mean 20,266 inhabitants, standard deviation 13,391, median 18,457 inhabitants, first quartile Q1—10,554, third quartile Q3—27,529), the population density per square kilometer ranges from 0.31–34,590.58 (mean 3486.36, standard deviation 6719.23, median 309.18, Q1 44.83, Q3 3752.54) [3].

Our objective in this paper is to introduce an online interactive application that helps in real-time monitoring of the COVID-19 pandemic in Catalonia, Spain. Since the outbreak of the pandemic, real-time interactive web applications have gained attention, especially in the domain of monitoring and modeling the dynamics of COVID-19. Throughout the globe several Shiny dashboards are currently deployed and managed by individuals, as well as government health organizations. Mukhtar et al. in their recent study explored and analyzed a series of novel and interesting web-based applications that have been specifically developed during the pandemic that can act as tools for the health professional community to help in advancing their analysis and research [5]. For example, Fernandez-Lozano et al. created an interactive dashboard to visualize all data related to the pandemic (cases, hospitalizations, and deaths) and its temporal evolution [6]. However, their tool does not display estimates or other variables (environmental, vaccine status, etc.). The app created by Galván-Tejada et al. offers greater interactivity and completeness than the former but falls short of reporting updated vaccination data as well as other variables, even though they provide valuable demographic information on top of a very thorough analysis of the pandemic in Mexico [7]. Another useful online web application for updated country-specific analysis and visualization is “COVID19-World” by Tebé et al. used for basic epidemiological surveillance covering time trends and projections, population fatality rate, case fatality rate, and basic reproduction number [8]. Wissel et al. developed “COVID-19 Watcher”, a similar web resource that displays COVID-19 data from every county and 188 metropolitan areas in the United States [9]. It provides the rankings of the worst-affected areas along with auto-generating plots depicting temporal changes in testing capacity, cases, and deaths. It is important to mention the “covid19.Explorer” R package and web application by Revell that has been designed to explore and analyze United States COVID-19 infection, death, and relative risk for different age groups with emphasis on geographic progress of the pandemic and effectiveness of lockdowns [10]. A similar interactive Python-based analytical tool to compare data and monitor trends across geographical areas related to the COVID-19 pandemic across counties in the United States and worldwide was developed by Zohner et al. [11]. “Mortality Tracker” is another interesting in-browser application developed by Almeida et al. mainly focused on the visualization of public time series of COVID-19 mortality in the United States [12]. It was developed in response to requests by epidemiologists to access the effect of COVID-19 on other causes of death by comparing 2020 real time values with observations from the same week in the previous 5 years, thus facilitating modeling of the interdependence between its causes. The literature shows a similar web application named “COVID19-Tracker” for Spain developed by Tobías et al. [13]. It produces daily updated data visualization and analysis of the COVID-19 diagnosed cases, and mortality in Spain. It also explores several analyses to estimate the case fatality rate, assessing the impact of lockdown measures on incident data patterns, estimating infection time and the fundamental reproduction number, and analyzing the mortality excess. An attempt at real-time statistical analysis in a user-friendly dashboard for researchers as well as the general public is made by Salehi et al. [14]. It includes two mathematical methods (pandemic logistic and Gompertz growth models) to predict the dynamics of COVID-19, as well as the Moran’s index metric, which provides a geographical perspective via heat maps and can help in the identification of latent reactions and behavioral patterns. Literature shows similar applications being implemented and maintained by researchers from various domains and different countries [15,16,17,18].

All these apps for respective regions systematically produce daily updated COVID-19 data visualizations and analysis. But a robust app with a combination of relevant socioeconomic and environmental risk factors and their interrelation with the dynamics of the pandemic has been less explored. Thus, this complete project represents eight interactive dashboards which collectively enable monitoring of the pandemic in Catalonia (PandemonCAT) and explore environmental and socioeconomic factors in the spatiotemporal evolution of the pandemic.

This is a multicenter project, led by the Research Group on Statistics, Econometrics and Health (GRECS) of the University of Girona, Spain, in which the Andalusian School of Public Health (EASP) (Granada, Spain) and the University of Granada also participate. The aim of the PandemonCAT project is to provide a web application that allows the monitoring of the COVID-19 pandemic in Catalonia. Its results include, in addition to vaccination, the results of diagnostic tests, transmission (reproductive number), hospitalization, ICU admissions, and the number of deaths. It also provides a visualization of those variables that could influence the spatiotemporal dynamics of the pandemic. Thus, in an interactive interface, the environmental variables (air pollutants and meteorological variables), the socio-economic variables, the points of interest where there may be a greater risk and the data of public mobility are shown.

The principal aim of the project is to provide a web application that allows the monitoring of the COVID-19 pandemic in Catalonia. Its results include, in addition to vaccination, the results of diagnostic tests, transmission (reproductive number), hospitalization, ICU admissions, and the number of deaths. It also provides a visualization of those variables that could influence the spatiotemporal dynamics of the pandemic. Thus, in an interactive interface, environmental variables (air pollutants and meteorological variables), socioeconomic variables, points of interest where there may be a greater risk, and data of public mobility are shown. The interactive web application reports a comprehensive list of all key variables with respect to the disease: new cases, hospitalizations, ICU’s, and deaths. We have also depicted the vaccination flow (both first and second doses) for different age groups of the population in individual health zones of the community. Finally, we report the smoothed standardized COVID-19 infected cases and mortality rates on maps of ABSs and regions of Catalonia. Table A1 in Appendix A reports information about individual dashboards of the application along with respective components and brief descriptions.

The rest of the article is organized as follows. In Section 2 we present an overview of the methodology followed to design PandemonCAT. The subsections report detailed descriptions of individual components of the complete methodology. Section 3 is devoted to presenting the results of individual components of PandemonCAT. In Section 4 we briefly discuss possible implications in other fields of study, as well as enhancements that may be implemented to further develop the current study. The article ends with a conclusion in Section 5.

## 2. Methods

PandemonCAT has been developed in the RStudio Shiny framework [19]. The application uses R packages to execute all analysis and plots internally. The key R packages used in the tool implementation include dplyr [20] and tidyverse [21] for data management. Packages like rgdal [22], sf [23], raster [24], maptools [25], and flowmap.blue [26] are used for spatial data analysis and visualizations. Interactive charts are generated with plotly package [27], while static graphical displays are designed using ggplot2 package [28]. Leaflet [29] along with leafpop [30] packages are used to generate interactive geospatial maps. The Shiny package [19] with Shiny flexdashboard [31] and rmarkdown [32] are used extensively for application enhancement and implementation as interactive web apps directly from R.

Figure 2 depicts the workflow diagram of the PandemonCAT application. Since 1 January 2020, we have retrieved, and curated data and it is being updated weekly with the new data reported by different data sources. The raw data accessed from multiple open data sources (referred in Section 2.1) are initially cleaned and preprocessed to ensure consistency and reliability. To speed up the analysis and visualization process, initial data wrangling techniques such as merging multiple heterogeneous data sources and discarding redundant variables and duplicate observations are performed. We automated the weekly data wrangling process because the raw datasets are extremely large and unstructured. The local databases are updated automatically every week in the dedicated cloud server. Section 2.2 provides a complete outline of the data remediation process. In the next phase, spatial, temporal, and spatiotemporal analysis are performed on the periodically updated datasets. All the analyses have been carried out using R version 4.0.1 [33]. The results of these analyses are displayed in interactive Shiny dashboards. Finally, all these dashboards are combined as a single application with brief information about each dashboard and individual access links. The integrated application runs on shared cloud servers shinyapps.io [34] that are operated by RStudio [35]. The site is maintained by the Research Group on Statistics, Econometrics and Health (GRECS), at the University of Girona, Spain.

### 2.1. Data Collection

All data displayed in PandemonCAT comes directly from official government sources. Specifically, we leverage the existence of open datasets as part of the increasing effort of the government to become more transparent. Early on during the COVID-19 pandemic, new datasets were created and shared, starting with the spatiotemporal incidence of both the number of cases and deaths by region and health zones (ABS). The existence of daily counts per day and geographic delimitation made a spatiotemporal representation possible. Unlike many autonomous communities, which only shared aggregated data (i.e., total counts per region, without the temporal evolution), Catalonia stood out in this regard and made apps such as PandemonCAT possible early on.

As the lockdown was ending (June 2020), more datasets were readily available such as the number of tests performed, the number of hospitalized, ICU and in the beginning of 2021, the number of vaccinated individuals. This allowed PandemonCAT to add all relevant variables to be displayed in our visualizations.

The current project uses daily updates from the following open COVID-19 datasets:

Number of:

Cases per ABS

Hospitalizations (and hospitalized) per region

ICU per region

Test per ABS

Vaccinated individuals per ABS

Official data sources are as follows:

General link with all the open datasets provided from the government: https://analisi.transparenciacatalunya.cat/ (accessed on 11 April 2022).

Regions: https://analisi.transparenciacatalunya.cat/api/views/c7sd-zy9j/rows.csv?accessType=DOWNLOAD&sorting=true (accessed on 11 April 2022).

ABS:

https://analisi.transparenciacatalunya.cat/api/views/xuwf-dxjd/rows.csv?accessType=DOWNLOAD&sorting=true (accessed on 11 April 2022).

Digitized cartography of the ABS:

Department de Salut. Cartography

https://salutweb.gencat.cat/ca/el_departament/estadistiques_sanitaries/cartografia/ (accessed on 11 April 2022)

Vaccine:

https://analisi.transparenciacatalunya.cat/api/views/tp23-dey4/rows.csv?accessType=DOWNLOAD&sorting=true (accessed on 11 April 2022).

Regarding the meteorological and air pollutant variables, the same official government datasets have been used. Due to their nature, they are not updated daily. Data shown in PandemonCAT is limited to the 2020 period, coinciding with the onset of the pandemic.

Meteorological variables:

METEOCAT, Generalitat de Catalunya, Meteorological data from XEMA

https://analisi.transparenciacatalunya.cat/Medi-Ambient/Dades-meteorol-giques-de-la-XEMA/nzvn-apee (accessed on 11 April 2022).

Air pollution:

https://analisi.transparenciacatalunya.cat/en/Medi-Ambient/Qualitat-de-l-aire-als-punts-de-mesurament-autom-t/tasf-thgu (accessed on 11 April 2022).

Socioeconomic variables:

We have used various sources for the socioeconomic variables: total population, percentage of population 65 years or more, percentage of population 0–25 years, and percentage of foreigners in 2020 from countries with medium and low human development index [2,36]

https://www.ine.es/dyngs/INEbase/en/operacion.htm?c=Estadistica_C&cid=1254736177012&menu=resultados&secc=1254736195461&idp=1254734710990#!tabs-1254736195557 (accessed on 11 April 2022).

Average income per person (in Euros):

Average of the years 2015, 2016, 2017 and 2018 [37]

https://www.ine.es/en/experimental/atlas/exp_atlas_tab_en.htm (accessed on 11 April 2022).

Unemployment rate [38]:

http://www.ine.es/censos2011_datos/cen11_datos_resultados_seccen.htm (accessed on 11 April 2022).

As in the case of pollutants, the data is limited to 2020, unless otherwise stated (for example, average income per person and unemployment rate).

### 2.2. Data Settings

Open data is an excellent source of information; however, raw data cannot be directly represented to convey important information regarding the state of the pandemic. Specifically, all figures need to be adjusted by population size (i.e., representing figures by 100,000 inhabitants). This adjustment is made possible due to the existence of updated demographic datasets with a low level of aggregation (for both regions and ABSs). The combination of both datasets (mainly thanks to the dplyr package [20] in R) has allowed us to create the following variables, which are the absolute reference for assessing the pandemic:

Weekly cases per 100,000 inhabitants: New cases adjusted for population on a 7-day window period. A new case is defined as those people who have received at least one positive PCR or antigenic test result during that period. The new case is allocated to the place of residence of such a person. That is if a person is registered to live in Barcelona, for instance, but gets a positive result from a hospital in other municipality, the new case is attributed to Barcelona.

Empiric 7-day Rt: indicates the rate of change of the new cases and is calculated as the ratio of the cumulative sum of weekly cases between t and t − 5.

Weekly tests per 100,000 inhabitants: PCR and antigenic tests performed on a 7-day period, adjusted for population, regardless of their results.

Weekly deaths per 100,000 inhabitants: new deaths attributed to COVID-19, adjusted for population, on a 7-day period.

Currently hospitalized per 100,000 inhabitants: number of people currently hospitalized due to COVID-19, adjusted for population.

Currently ICU per 100,000 inhabitants: number of people currently in intensive care unit (ICU) due to COVID-19, adjusted for population.

On the vaccination front, the same procedure of adjusting by population is performed at the health zone level. This has proved to be critical, especially since the inter-health zone population differences are large (and thus, absolute numbers don’t give an accurate picture of the progress of the vaccination campaign).

### 2.3. Spatial Prediction of Air Pollutant Levels

In this section we provide the spatial predictions of the levels of atmospheric pollutants for each ABS in Catalonia. The problem is that the air pollution monitoring stations are not distributed homogeneously throughout the territory of Catalonia, but rather are concentrated, mainly in the Barcelona region. Therefore, we follow our previous work [39].

Specifically, our objective there was to perform spatial predictions of air pollution levels using a hierarchical Bayesian spatiotemporal model [39,40] that allowed us to perform the predictions in an effective way and with very few computational costs [39]. We used the Stochastic Partial Differential Equations (SPDE) representation [41] of the integrated nested Laplace approximations (INLA) approach [42,43] to spatially predict, in the territory of Catalonia, the levels of the four pollutants for which there is the most evidence of an adverse health effect: coarse particles, nitrogen dioxide, ozone, and carbon monoxide (pollutants of interest) [39]. We performed the spatial predictions at a point level (defined by its UTM coordinates), allowing them to be aggregated later in any spatial unit required (ABSs in our case). We were especially interested in the long-term exposure to air pollutants. That is, by living in a certain area an individual is exposed to a mix of pollutants that have lasting effects on their health.

We obtained information on the levels of air pollutants for 2011–2020 from the 143 monitoring stations from the Catalan Network for Pollution Control and Prevention (XVPCA) (open data) [44], located throughout Catalonia and that were active during that period. The pollutants we were interested in for making spatial predictions were coarse particles, those with a diameter of 10 μm (μm) or less (PM_10_), nitrogen dioxide (NO_2_), ozone (O_3_) (all of them expressed as μm/m^3^) and carbon monoxide, CO (all of them expressed as mg/m^3^) (air pollutants of interest, hereinafter). However, the monitoring stations also measured other pollutants: fine particles, those with a diameter of 2.5 μm or less (PM_2.5_), nitrogen monoxide (NO), sulphur dioxide (SO_2_), benzene (C_6_H_6_), hydrogen sulphide (H_2_S), dichloride (Cl_2_), and heavy metals (mercury, arsenic, nickel, cadmium, and lead).

We specified a hierarchical spatiotemporal model:$$Z\left(s_{i},t\right)=Y\left(s_{i},t\right)+\varepsilon \left(s_{i},t\right)$$
where *i* denotes the air pollution monitoring station where the pollutant was observed; *t* is the time unit; $s_{i}$ the location of the station; Y(.,.) the spatiotemporal process, the realization of which corresponds to the pollutant measurements (at station *i* and time unit *t*); and ε(.,.) the measurement error defined by a Gaussian white-noise process (i.e., spatially and temporally uncorrelated).

The spatiotemporal process, Y(.,.), is a spatiotemporal Gaussian field that changes in time according to an autoregressive of order one (AR(1)).

The measurement equation was specified as:$$y\left(s_{i},t\right)=\mu \left(s_{i},t\right)+\eta \left(s_{i},t\right)$$
where *μ*(.,.), denotes a large scale component and *η*(.,.) the realization of a spatiotemporal process, specified as,
$$\eta \left(s_{i},t\right)=\phi \eta \left(s_{i},t-1\right)+\omega \left(s_{i},t\right)\;\mathrm{where}\;\left|\phi \right|<1.$$

$\omega \left(s_{i},t\right)$, which was assumed to be a zero mean Gaussian and a Matérn covariance function:$$Cov\left(\eta \left(s_{i},t\right),\eta \left(s_{i}^{\prime },t\right)\right)=\frac{\sigma^{2}}{2^{\nu -1}\Gamma \left(\nu \right)}\;{\left(\kappa \|s_{i}-s_{i}^{\prime }\|\right)}^{\nu }{\;Κ}_{\nu }\;\left(\kappa \|s_{i}-s_{i}^{\prime }\|\right)$$
where ${Κ}_{\nu }$ is the modified Bessel function of the second type and order ν>0, ν is a parameter controlling the smoothness of the GF, $\sigma^{2}$ is the variance, and κ>0 is a scaling parameter related to the range, ρ, the distance to which the spatial correlation becomes small.

The linear predictor of the GLMM specification of the large-scale component, μ(.,.) was,
$$\mu_{i,t}=\beta_{0}+\sum_{j=1}^{14}\beta_{j}pollutant_{j,it}+\beta_{15}altitude_{i}+\beta_{16}area_{i}+\eta_{i}+S_{i}+\tau_{month}+\tau_{year}$$

We included as covariates: (1) *pollutant*: the pollutants of interest other than the pollutant for which the spatial prediction was made and, second, the rest of the pollutants that are measured in each monitoring station (i.e., PM_2.5_, NO, SO_2_, C_6_H_6_, H_2_S, Cl_2_, mercury, arsenic, nickel, cadmium, and lead); (2) *altitude*: the altitude of the air pollution monitoring station; and (3) *area:* the area of the ABS. On the other hand, including random effects, we controlled for heterogeneity (those unobservable factors that could be associated with the levels of the pollutant) $\eta_{i}$ (unstructured random effect indexed on the ABS), spatial dependence (that is, the existence of geographic patterns), $S_{i}$ (structured random effect according a Matérn); and temporal dependence (trend and seasonality), $\tau_{year}$ and $\tau_{month}$, respectively (structured random effects indexed on year and month, respectively).

### 2.4. Smoothing of the Rates of the Outcomes from COVID-19

The simplest disease (or mortality) maps represent the cases or deaths observed in each geographic area. However, any interpretation of the geographical structure of the cases is limited by the lack of information on the spatial distribution of the population that could be ‘at risk’ and, consequently, has generated these observed cases. Therefore, the representation of rates that allow incorporating the effect of the population at risk is preferred, instead of representing gross cases. However, the direct use of crude rates does not allow comparison between different areas, since the differences observed between them may be due to risk factors that have not been considered, such as age. One measure that considers the age structure is the age-standardized rate. There are two methods for age standardization, which are known as direct and indirect standardization. In the representation of disease maps, the use of the indirect method is preferred, which consists of comparing the observed cases of the disease in an area with the expected cases if the risks for each age group were the same as in a certain area reference population. The observed/expected ratio is called the standardized incidence (or mortality) rate (SIR or SMR), which is nothing more than an estimator of the relative risk of the area, that is, of the risk of illness (or death) in relation to the reference group [45,46].

SIRs (or SMRs), even though they have been widely used, have some limitations. They depend to a great extent on the population size, since the variance of the standardized rates is inversely proportional to the expected values; thus, areas with little population will present estimators with great variability. In this sense, the extreme standardized rates and, therefore, dominant in the apparent geographic pattern, are those estimated with the least precision in areas with few cases. In addition, the variability of the observed cases is usually much greater than expected, producing what is called ‘extra variability’. In fact, when spatial data are available it is important to distinguish two sources of extra variation. In the first place, the most important source is usually the so-called ‘spatial dependence’, which is a consequence of the correlation of the spatial unit with neighboring spatial units, generally those that are geographically contiguous. Thus, the standardized rates of contiguous, or close, areas are more similar than the standardized rates of spatially distant areas. Part of this dependency is not really a structural dependency but is mainly due to the existence of uncontrolled variables, i.e., those not included in the analysis. Regarding the second source, the existence of extra independent and spatially unrelated variation, called ‘heterogeneity’, due to unobserved variables without spatial structure that could influence the risk must be assumed [45,46].

To solve the problems derived from the direct use of SIRs (or SMRs), several alternatives have been proposed to “smooth” them, that is, to reduce the extra variation. Specifically, to estimate disease risks it is preferable to use models (known as ‘disease mapping models’) since they allow incorporating explanatory variables and borrowing information from neighboring areas to improve local estimators, smoothing the extreme values because of small sample sizes [45,46].

Here, to smooth the SIRs, we used a log Gaussian Cox (LGCP) model. The LGCP model is the analogue of the Gaussian linear model used for geostatistical data when data is modelled in the form of point processes. However, this model is currently being used to approximate spatial data of any type (that is, areal data, geostatistical data, and point processes) [47].

First, we assessed the existence of a geographic pattern, as well as clusters of cases in the incidence and mortality of COVID-10. To do this, we specified an LGCP, in which we did not include explanatory variables but only random effects that captured: (1) individual heterogeneity not spatially structured, that is, it collects those unobservable confounders associated with each ABS that do not vary over time; (2) the time trend of the risk (in a non-linear way); and (3) the spatial dependence. In our case, the LGCP model had three peculiarities. First, we included as an offset (denominator) the expected number of cases and deaths from COVID-19 in the ABS. In this way we smoothed the SIRs. Second, since there were ABSs that some weeks did not have any cases or deaths, we allowed the dependent variables to have an excess of zeros, assuming that they are distributed according to a negative binomial. Third, in addition to controlling for heterogeneity, spatial dependence, and temporal dependence using random effects, we allowed the spatial pattern of incidence and mortality to vary over time, including a random effect for the interaction between the spatial and the temporal components [48].

Second, in the previous model we included those variables that could have explained the risk of incidence and mortality and, therefore, also the possible geographic patterns and the existence of clusters, if any. As explanatory variables we included socioeconomic variables net income per person (average 2015 to 2018), unemployment rate, population density, percentage of the population aged 65 years or more (average 2015 to 2018), percentage of slums (with surface area smaller than 40 m^2^), and percentage of residents born in low-income countries); meteorological variables (net effective temperature—a thermal index that combines temperature, relative humidity and wind speed—and atmospheric pressure); long-term exposure (from 2009 to 2019) to air pollutants (PM_10_, NO_2_, and O_3_); mobility variables (exits, entrances, and internal movements) and the accumulated weekly percentage of those vaccinated with two doses. All variables were included at the ABS level. The socioeconomic variables (except for density) were collected at the census tract level and for this reason the values for each ABS were obtained by means of a weighted average of the census tracts contained in them, using the ABS population as weights. The values of the meteorological variables and atmospheric pollutants in each ABS were spatially predicted using a hierarchical Bayesian spatiotemporal model. With the exception of the accumulated percentage of those vaccinated with two doses, the rest of the time-dependent variables, that is, for which we had weekly values (meteorological, atmospheric pollutants, and mobility) were included in the model as the average of the values of the previous two weeks (since their effect on incidence and mortality, if any, was not immediate). Finally, we allowed the relationship between incidence and mortality and the explanatory variables to be non-linear.

## 3. Results

The spatiotemporal and visual analytics capabilities included in PandemonCAT can be useful to explore associations and trends among meteorological and air quality variables and COVID-19 indicators, with a layer of socioeconomic and public mobility information. The app demonstrates visualization of these variables that could have some influence on the spatiotemporal dynamics of the pandemic. We believe the addition of these new dimensions, which is the key innovation of our project, will improve the current understanding of the spread and impact of COVID-19.

The application can be accessed online (https://www.udg.edu/pandemoncat, (accessed on 11 April 2022)). The application is a dynamic and interactive dashboard, as illustrated in Figure 3, which allows the user to get an overview of the entire application and its different modules. The user can get detailed information and links for the component modules by clicking on individual tabs.

Figure 4 shows the information tab for one component module of PandemonCAT. It provides an outline concept of the particular dashboard and its unique functionalities along with data type and sources. It also provides the weblink of the component module as highlighted in Figure 4.

In the following sections we demonstrate the characteristics of individual components of PandemonCAT through various figures supported by relevant functional explanations.

### 3.1. COVID-19 Dynamics

The “COVID-19 Dynamics” app depicts the principal parameters to track the COVID-19 pandemic in Catalonia (Figure 5). In the first section, we present a spatiotemporal map at the ABS level which is the lowest administrative aggregation level for data collection from official open data portals. The interactive map has the option to display the results categorized by the parameters, namely, empirical 7-day reproductive number and weekly records per 100,000 inhabitants for infected cases, tests performed, and deaths. Moreover, it also provides options to check the currently hospitalized and intensive care unit (ICU) patients per 100,000 inhabitants for individual ABSs. Details of these variables are discussed in Section 2.2. The app also provides the option to select any particular date starting from 20 March 2020 to explore spatial distribution of any variables mentioned above. In the next section, a time plot for each of the same variables is available to review its evolution for individual ABS compared with Catalonia as a whole. In this section, the user will have the option to select a particular time period mentioning a start and end date. It is worth noting that, though the majority of the populations are between 20,000 and 40,000, there exists a particular heterogeneity in the population size of each ABS. This fact may be relevant since outliers are more often found in ABSs with low populations that experience serious outbreaks. Figure 5 (left) depicts the parameter options to control the spatiotemporal visualizations. Right (top) map shows the spatial variation of weekly infected cases per 100,000 population for a particular selected date in different ABSs of Catalonia. While the plot on right (bottom) presents the temporal trend of the same variable for a particular ABS compared with the entirety of Catalonia for a selected range of time. In both map and linear plots, the user can get detailed information for any spatial and temporal resolution with a click (as shown on the map pop-up information window).

### 3.2. Vaccine Rates

The “Vaccination” app displays the progress being made in the vaccination campaign against COVID-19 in Catalonia (Figure 6).

In the first section, we present a spatiotemporal map at the ABSs level representing the share of population with either one or two doses of all types of vaccine, for both cumulative and the weekly proportion. In addition to the overall share of population, specific shares per age group can also be explored. The age groups are all 10-year periods, except for “80 or above” and “0 to 19” years old. In the second section, a time plot of the same variables is available to monitor its evolution. The plot for Catalonia is also displayed to provide some context for the specific ABS plot. Note that, similar issues related to outliers found in ABSs with low populations (also mentioned in Section 3.1) can be observed due to heterogeneity in the population size of each ABS. Figure 6 (left) depicts the parameter options to control the spatiotemporal visualizations. Right (top) map shows the spatial variation of cumulative vaccination for the population of all age groups on a particular selected date in different ABSs of Catalonia. This map depicts the distribution for all types of vaccines for first dose only. While the plot on the right (bottom) presents the temporal trend of the same variable for a particular ABS compared with the entire Catalonia. In both map and linear plots, the user can get detailed information for any spatial and temporal resolution with a click (as shown on the map pop-up information window).

### 3.3. Meteorological Variables

The “Meteorological” app displays daily average records of six meteorological components for individual weather stations in Catalonia (Figure 7). The six components included in the current project are atmospheric pressure, precipitation, relative humidity, solar irradiance, temperature, and wind velocity.

In the first section, we present the spatial distribution of daily average meteorological records from 75 weather stations located in different ABSs of Catalonia. The second section depicts a smoothed scatter plot of average monthly records of individual meteorological components for all ABSs of Catalonia. Figure 7 (left) shows the option to select the type of meteorological components and the date slider. The right section top of Figure 7 presents the map of Catalonia with the locations of individual weather stations. Clicking on any station displays detailed values of the weather component on the selected day for that particular station. A smoothed scatter plot is displayed below.

### 3.4. Air Pollutants

This app focuses specifically on the daily average concentration of coarse particles (PM_10_), nitrogen dioxide (NO_2_), ozone (O_3_), and carbon monoxide (CO) recorded at 75 pollution monitoring stations in the region (Figure 8).

In the first map we report the daily average concentration of air pollutants for the pollution monitoring stations distributed over individual ABSs of Catalonia. The smoothed scatter plot below displays the overall behavior as a monthly average of individual air pollutants for all ABSs of Catalonia. The second map for all ABSs depicts the average annual prediction for a given air pollutant concentration. The prediction map provides an option to select any year between 2011 and 2020, inclusive. Figure 8 (left) shows the options to select the type of air pollutants and the date slider along with the variations of selected pollutants for different pollution recording stations. Below it displays a smoothed scatter plot. Figure 8 (right) presents the spatial distribution of average annual prediction for the average annual concentration of a selected pollutant and for a selected year.

### 3.5. Socioeconomic Variables

The first section of this app reports the spatial distribution of variables such as income per capita, percentage of population (65 or more), percentage of population (0–25), percentage of foreign population, and unemployment rate (Figure 9). The spatial resolution of the variables is ABSs of Catalonia and it reflects substantial inequalities across regions.

The second map displays the geographic locations of points of interest (POIs), weather stations and pollution recording stations in an interactive map interface (Figure 10). POIs are potential COVID-19 contamination hotspots like restaurants, night clubs, bars, and other similar public aggregation hotspots. Figure 9 displays the spatial distribution of total population in the 343 ABSs of Catalonia. The map depicts a wide heterogeneity of population in the entire region. On the other hand, Figure 10 displays clustered POIs distributed over the entire study region.

### 3.6. Public Mobility

In Catalonia during late February 2020 and early March 2020, there were no strong actions or precautions taken by the government warning of the seriousness of the pandemic. Community transmission started in mid-February and by 13 March, confirmed cases of COVID-19 had been recorded in almost all the 343 ABSs of the region. This led to the implementation of nation-wide lockdown in Spain on 14 March 2020 which was also effective in Catalonia. The lockdown continued for more than 3 months. In the beginning of June 2020 with a gradually decreasing trend in the number of infected and deaths, the government started lifting some restrictions and relaxing the lockdown to some extent. Leveraging the recently available public mobility open data, in the Public Mobility tab we are able to provide exact figures for every municipality in Catalonia, including long trips and shorter ones (such as the daily commute to work).

This new dimension is key for understanding and quantifying the impact of non-pharmaceutical interventions (NPIs) throughout the pandemic. Never before has an open dataset provided so much insight into the daily mobility of the population, and thanks to it, we can easily spot the stay-at-home period of March–April 2020 in comparison with the following months, proving once again the high degree of compliance with that specific NPI (Figure 11). The dynamic flow map in Figure 11 represents average inter- and intra-ABS public mobility during October 2020. The user has the option to select the months from March to November 2020 which covers all the phases—before, during, and after the lockdown period.

### 3.7. Smoothing

We show the weekly smoothed standardized incidence (positive cases) and mortality rates by COVID-19 on maps of Catalonia by ABS and region (Figure 12). These smoothed rates allow the user to glimpse the existence of geographic patterns in incidence and mortality.

Those ABSs with smoothed rates higher than unity have a risk of incidence higher than expected and those with smoothed rates lower than the unit, a lower risk than expected.

To help evaluate the existence of agglomerations of excess cases (i.e., clusters), we also show the ‘exceedance probability’, which is the probability that the smoothed rate is above 1. Those ABSs or regions with a probability greater than 80% can be classified as high risk (of incidence or mortality) and those with a probability less than 20% of low risk.

We show smoothed rates without including explanatory variables and adjusting for various explanatory variables. In the latter case, including only vaccination (weekly accumulated percentage of people vaccinated with the two doses) and including vaccination, socioeconomic and environmental variables (meteorological and air pollutants), and mobility.

#### 3.7.1. Weekly Positive Cases

Those ABSs with smoothed rates higher than unity have a risk of incidence higher than expected and those with smoothed rates lower than the unit, a lower risk than expected. For example, a smoothed incidence rate equal to 1.5 means that in that ABS there were 50% more cases than expected (according to the age and sex structure of the ABS).

Regarding the probabilities of exceedance, those ABSs with a probability greater than 80% can be classified as high risk of incidence and those with a probability less than 20% of low risk.

#### 3.7.2. Weekly Deaths

The smoothed rates allow the user to view geographic patterns in mortality. For example, regions with a smoothed mortality rate equal to 1.2 means that the number of deaths from COVID-19 was 20% higher than expected.

Regarding the probabilities of exceedance, those regions with a probability greater than 80% can be classified as high risk of mortality and those with a probability less than 20%, as low risk.

## 4. Discussion

It is important for the health professionals and policymakers to have access to the most relevant, reliable, and real-time information that can be used in their day-to-day tasks of COVID-19 research and analysis.

In this context, all apps referred to in Section 1 produce daily updated COVID-19 data visualizations and analyses. The results of our current study illustrate that, PandemonCAT is a novel interactive web application which acts as a collective monitoring package for daily COVID-19 updates along with regional vaccination flow and several environmental and socioeconomic variables that could have some influence on the spatiotemporal dynamics of the pandemic. The app explores variables such as meteorological and air pollution variables, population by age group, unemployment rate, income per capita, and others in a geospatial interface. It also provides an interactive interface to visualize public mobility before, during, and after the lockdown phases in the community. The visualization of these variables could have some influence on the spatiotemporal dynamics of the pandemic.

On the other hand, linking the pandemic severity with environmental factors such as air pollution, we find the article from Martorell-Marugán et al. which may be the closest to our study in that regard [49]. Combining great visualization capabilities with sound and rigorous statistical analysis of the possible contributing factors of the pandemic goes hand in hand with our aim. We tried to take it a step forward, albeit limiting it to Catalonia due to data availability, and relate many other relevant variables that enrich the overall picture. Related to the social contact data sharing initiative, an interactive tool (SOCRATES) to assess mitigation strategies for COVID-19 was developed by Willem et al. [50]. It implements location-specific physical distancing measures (e.g., schools or at work) and captures their impact on the transmission dynamics.

To the best of our knowledge, this is the most complete, open, and free source of public health information regarding the pandemic and its possible contributing factors in Catalonia. Many other applications have been developed throughout this period; however, none offer the end user the option to explore with a single click other key variables such as social mobility, meteorological and air pollution statistics. Another unique aspect of the current application is the level of spatial resolution. Limited online applications provide dynamic spatial resolution up to the level of ABSs. In addition, it is the only interactive application which provides a visualization of human mobility and highlights its influence on the transmission of COVID-19 for individual ABSs and regions of Catalonia. To explore the link between COVID-19 transmission and air pollution, PandemonCAT is one of the few online applications to provide spatial predictions of pollutants for the entire time frame which covers all the phases before, during, and after the lockdown periods. We also report that, to the best of our knowledge, no other application provides dynamic, smoothed, standardized COVID-19 infected cases and mortality rates. These smoothed rates allow the user to explore the existence of geographic patterns in incidence and mortality rates. Moreover, the datasets used as input for the application are collected from official open data portals. This makes the data collection process smoother and on the other hand allows additional individuals to analyze and interpret the data, making it transparent and reproducible. The entire application can be easily replicated using open data from any other region or country.

Although the sources of our data are very complete and informative, there are several limitations, which are common to many countries. First and foremost, during the first COVID-19 wave, testing for the disease was very limited, making it hard to estimate the true number of infected during that period. Data on hospitalizations and deaths was also constrained by the testing capacity, but it did so to a lesser degree since they were greatly prioritized. This fact is emphasized as total COVID-19 deaths in Catalonia account for almost all the excess deaths for the first epidemic wave.

However, this limitation became virtually eliminated during the month of July 2020, when the testing capacity was greatly expanded. This is reflected by much lower positive rates, even when the next several waves were at their highest, and by a much lower share of cases that ended up in hospitalization or death.

Another limitation is the lack of distinction between the type of vaccine being administered to the population. We do have information available regarding the total numbers administered for each manufacturer (either Pfizer/BioNTech, ModeRNA, Janssen or AstraZeneca), but given that it’s likely that each vaccine offers a different protection profile, especially with the expected waning immunity, we may have a certain degree of heterogeneity among the fully vaccinated cohort.

## 5. Conclusions

The complex nature of the COVID-19 epidemic and its dynamics of spread and transmission in the global population demands that researchers and health professionals embrace a multidisciplinary approach in addressing the challenges raised by the pandemic. Thus, it is essential to have efficient web-based applications or, portals that can provide the most relevant, reliable, and up-to-date information with a single click. In this context, the dynamic web-application we have developed offers a tool to scientists and others in the broader community to visualize the spatiotemporal trends of COVID-19 and enables comparisons at the ABS level in Catalonia, Spain. The features we incorporated in our open-source web application provide a comprehensive picture of public mobility, environmental, and other socioeconomic aspects that may have an impact on the spatiotemporal dynamics of the pandemic. The visualization of spatial predictions of pollutants related to COVID-19 is another novel feature of PandemonCAT. Finally, the interactive functionality to depict dynamic, smoothed, standardized COVID-19 infected cases and mortality rates help in providing an insight for the policymakers in developing public health strategies and control measures related to the ongoing pandemic.

## Figures and Tables

**Figure 1 ijerph-19-04783-f001:**
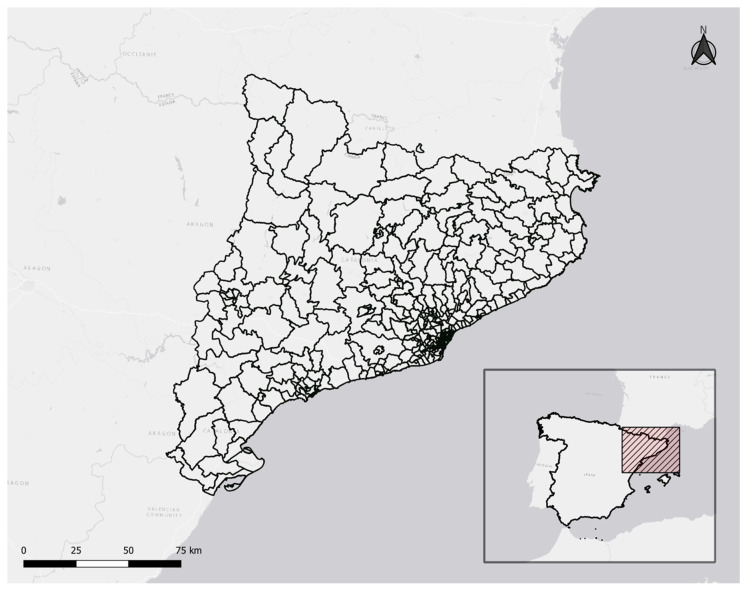
Geographic location of 343 ABSs of Catalonia.

**Figure 2 ijerph-19-04783-f002:**
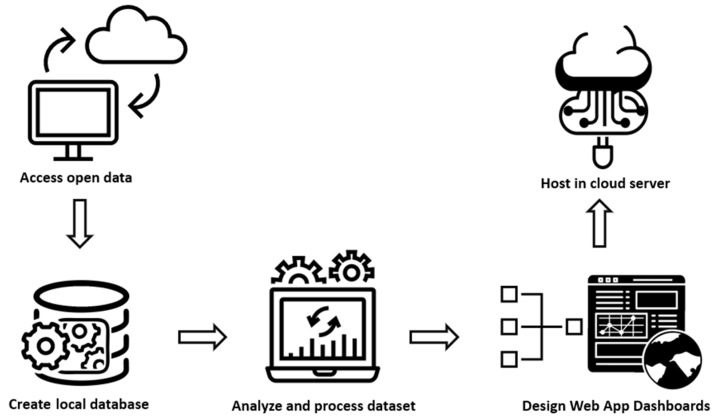
Workflow diagram of PandemonCAT.

**Figure 3 ijerph-19-04783-f003:**
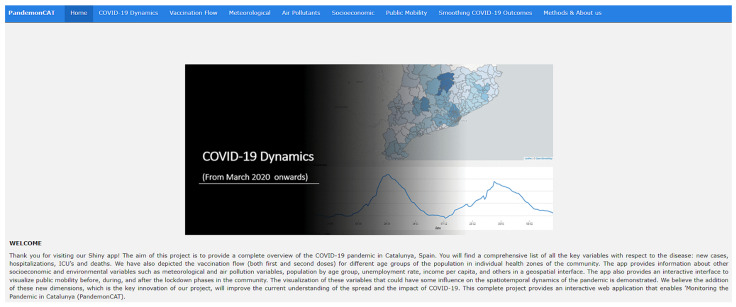
Overview of the PandemonCAT application.

**Figure 4 ijerph-19-04783-f004:**
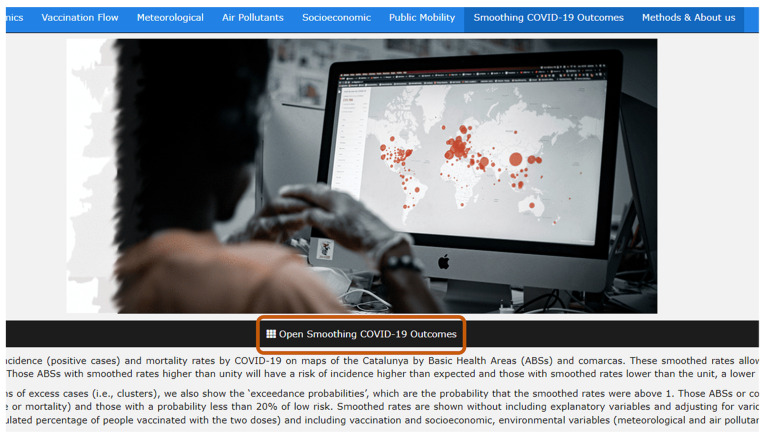
Dashboard for component module of PandemonCAT.

**Figure 5 ijerph-19-04783-f005:**
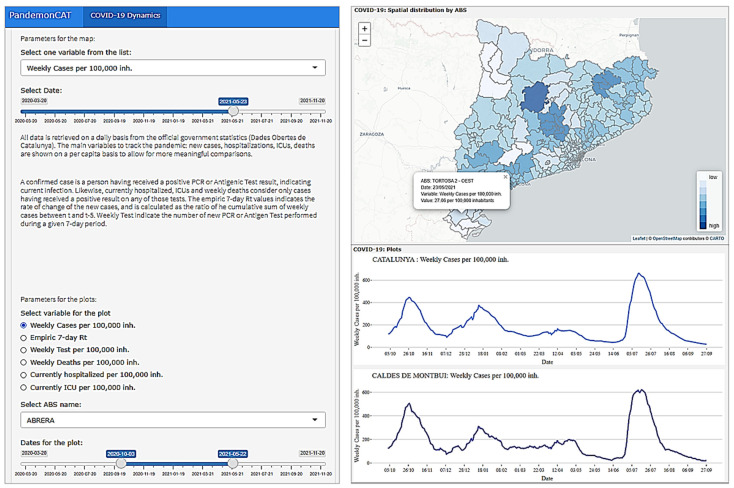
Dashboard of PandemonCAT that allows to analyze the spatiotemporal dynamics of COVID-19 in Catalonia.

**Figure 6 ijerph-19-04783-f006:**
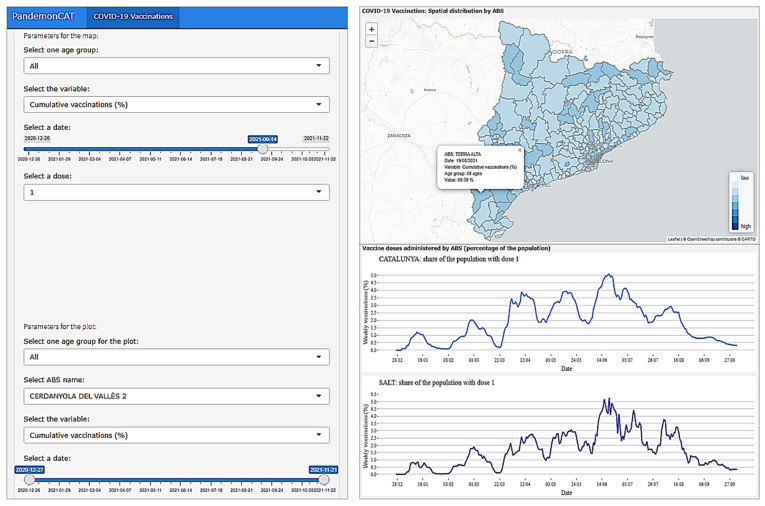
The dashboard of PandemonCAT enables analysis of the spatiotemporal dynamics of COVID-19 vaccination in Catalonia.

**Figure 7 ijerph-19-04783-f007:**
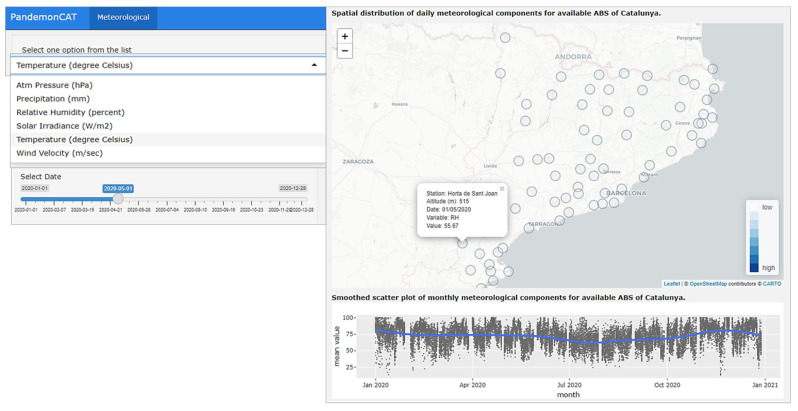
Dashboard of PandemonCAT displaying daily average records of meteorological components from different weather stations in Catalonia.

**Figure 8 ijerph-19-04783-f008:**
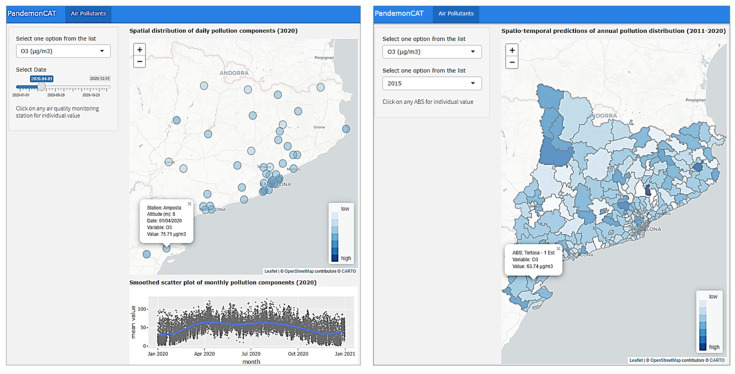
Dashboard of PandemonCAT that provides visualization of the spatiotemporal variation of principal air pollutants in Catalonia.

**Figure 9 ijerph-19-04783-f009:**
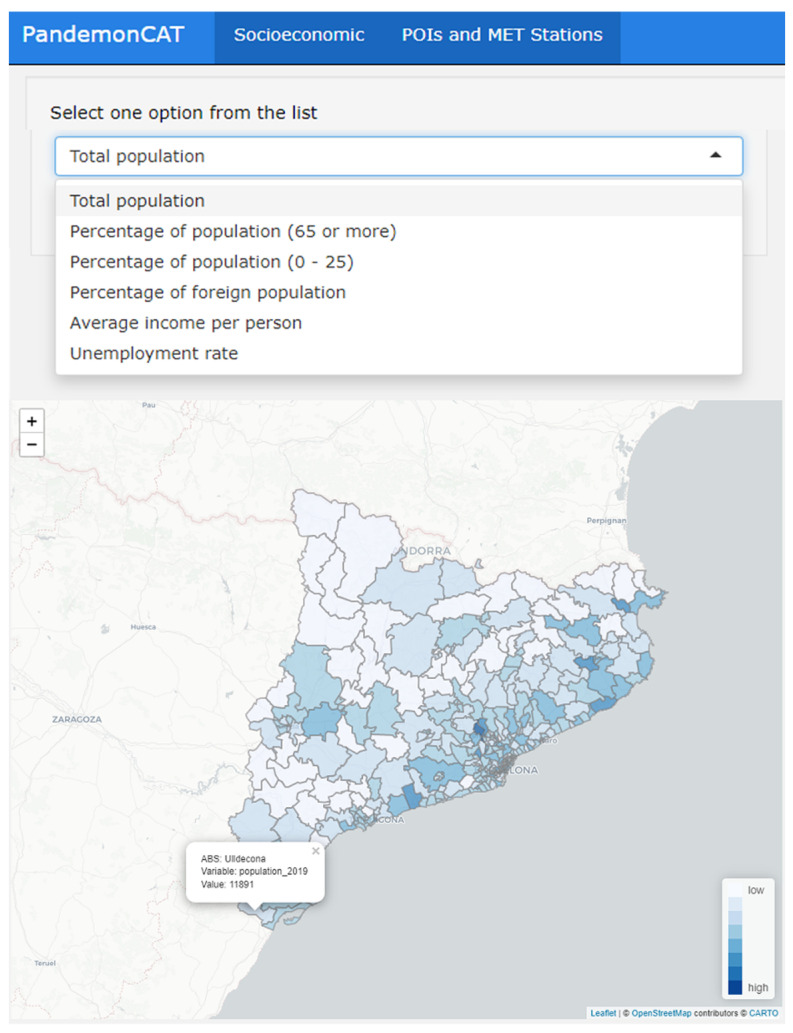
Dashboard of PandemonCAT displaying spatial distribution of socioeconomic variables.

**Figure 10 ijerph-19-04783-f010:**
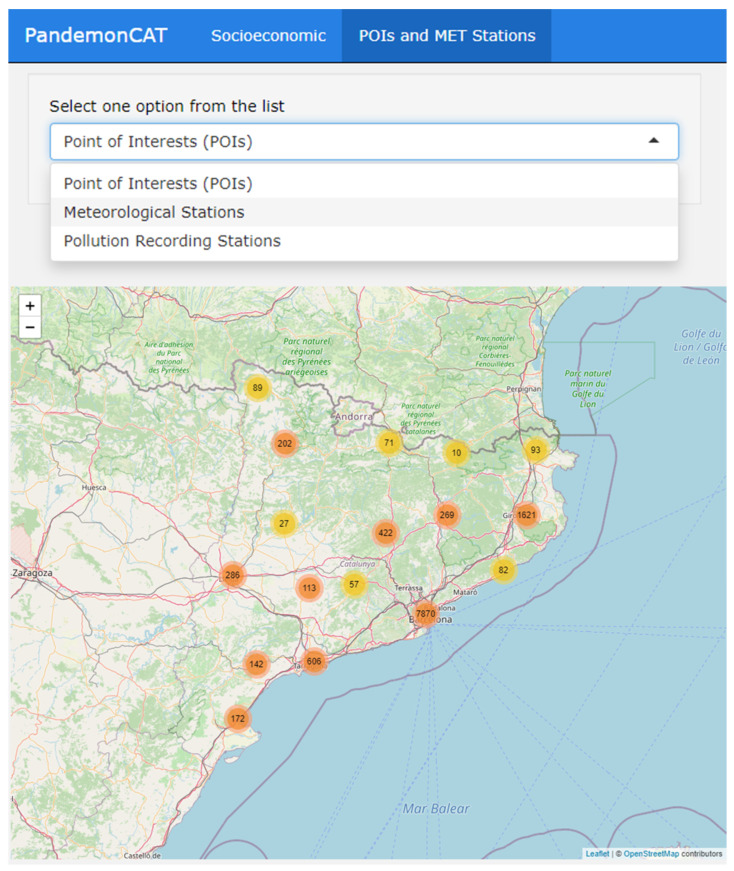
Dashboard of PandemonCAT displaying spatial distribution of POIs. The number indicated in the circle represent the number of units in the cluster.

**Figure 11 ijerph-19-04783-f011:**
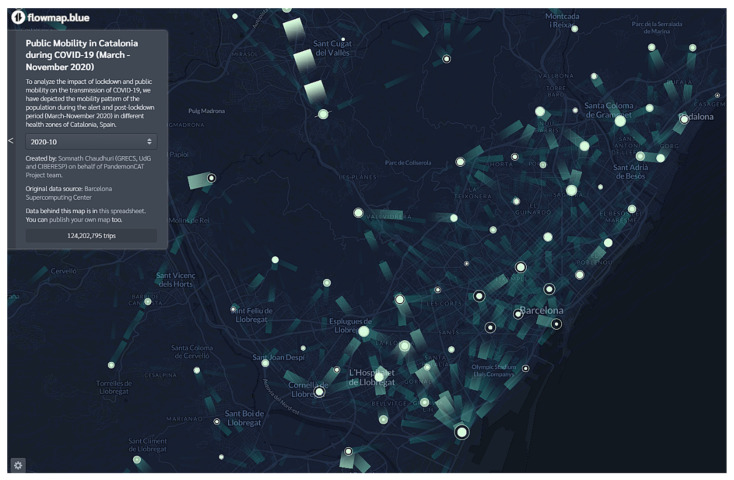
Dashboard of PandemonCAT to visualize inter- and intra-ABS monthly public mobility.

**Figure 12 ijerph-19-04783-f012:**
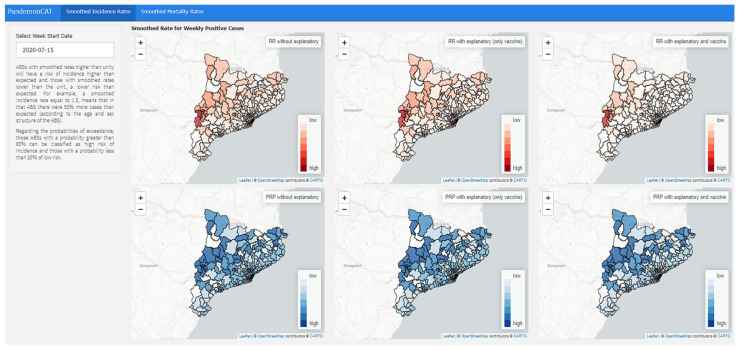
Dashboard of PandemonCAT that allows analysis of weekly smoothed standardized incidence (positive cases) of COVID-19.

## Data Availability

We used open data with free access. All data sources are referred in Section 2.1. Code will be available at www.researchprojects.es.

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
