# Peer review of "PandemonCAT: Monitoring the COVID-19 Pandemic in Catalonia, Spain"

_ijerph, 2022, doi:10.3390/ijerph19084783_

Round 1

Reviewer 1 Report

Reccomended for publication after modifications as suggested. 

Author Response

Reviewer #1

The authors of this study present an online interactive application that uses R software and the Shiny package to help in real-time "Monitoring COVID-19 Pandemic in Catalonia, Spain." The authorscompiled weekly COVID-19 dynamics updates in a geospatial interactive platform for particular Catalanbasic health zones. It also depicts the evolution of vaccination campaigns on a georeferenced map,with the fraction of the population receiving one or two vaccine doses for populations of various ages.In addition, the application includes an interactive interface to visualise monthly public mobility before, during, and after the lockdown periods, as well as information about environmental and socioeconomic elements. Finally, the authors provide smoothed standardised COVID-19 infectedcases and fatality rates on ABS and Catalan region maps. The user can utilise these smoothed rates to see if there are any geographic patterns in incidence and mortality rates. The authors also illustrated the visualisation of variables that could have an impact on the pandemic's spatiotemporal dynamics. Finally, the authors stated that the addition of these new dimensions, which is the project's maininnovation, will increase current knowledge of COVID-19's distribution and impact in the community. They also discovered that this programme can be utilised as an open tool for public consultation inCatalonia and throughout Spain. It could also have consequences in public health by making datavisualisation easier and enabling for timely interpretation due to the pandemic's unpredictable nature.

We thank the reviewer for their insightful comments.

Overall work is good, more related to the current pandemic situation and interesting. I recommendthis work for publication provided the following observations/modifications/corrections:

In section 2.3, the authors mentioned about stochastic partial differential equation representations ofintegrated nested Laplace approximations approach for spatial predict. The authors need to provide those mathematical equations with conditions to get more insight of the current in view of multidisciplinary. Also, authors need to mention Bayesian spatiotemporal model in terms ofmathematical equations to get more insight in view of other researchers.

The authors need to specify a measurement equation mathematically and which is the composition ofthe realization of the spatiotemporal process which corresponded to the pollutant measurements and a large-scale component, depending on the covariates to get more insight and realistic.

Authors need to provide more information about a spatiotemporal Gaussian process with zero meanand covariance function in terms of mathematical equations to get more insight.

How to control heterogeneity, spatial dependence and temporal dependence? Some more explanation is required.

All these comments we have tried to answer with a new wording of Section 2.3. On page 8,

‘2.3.- Spatial prediction of air pollutant levels

In this section we provide the spatial predictions of the levels of atmospheric pollutants for each of the ABS in Catalonia. The problem is that the air pollution monitoring stations are not distributed homogeneously throughout the territory of Catalonia, but rather are concentrated, mainly in the Barcelona region. Therefore, we follow our previous work [39].

Specifically, our objective there was to perform spatial predictions of air pollution levels using a hierarchical Bayesian spatiotemporal model [39,40] that allowed us to perform the predictions in an effective way and with very few computational costs [39]. We used (…)’

but especially on pages 9 and 10,

We specified a hierarchical spatiotemporal model:

where i denoted the air pollution monitoring station where the pollutant was observed; t was the time unit;  the location of the station;  the spatiotemporal process, the realization of which corresponded to the pollutant measurements (at station i and time unit t); and  the measurement error defined by a Gaussian white-noise process (i.e., spatially and temporally uncorrelated).

The spatiotemporal process, , was a spatiotemporal Gaussian field that changes in time according to an autoregressive of order one (AR(1)).

The measurement equation was specified as:

where , denotes a large scale component and  the realization of an spatiotemporal process, specified as,

       where .

 which was assumed to be a zero mean Gaussian and a Matérn covariance function:

where  is the modified Bessel function of the second type and order .  is a parameter controlling the smoothness of the GF,  is the variance and , is a scaling parameter related to the range,  the distance to which the spatial correlation becomes small.

The linear predictor of the GLMM specification of the large-scale component,  was,

+

We included as covariates: i) pollutant: the pollutants of interest other than the pollutant for which the spatial prediction was made and, second, the rest of the pollutants that are measured in each monitoring station (i.e., PM2.5, NO, SO2, C6H6, H2S, Cl2, mercury, arsenic, nickel, cadmium and lead); 2) altitude: the altitude of the air pollution monitoring station; and 3) area: the area of the ABS. On the other hand, including random effects, we controlled for heterogeneity (those unobservable factors that could be associated with the levels of the pollutant)  (unstructured random effect indexed on the ABS), spatial dependence (that is, the existence of geographic patterns),  (structured random effect according a Matérn); and temporal dependence (trend and seasonality),  and , respectiviely (structured random effects indexed on year and month, respectively).

How SIR depend to a great extent on the population size? Some relevant information needs to mention in the literature.

Please excuse us, but we have already replied to this comment in the manuscript. Specifically, on page 10,

‘SIRs (or SMRs), even though they have been widely used, have some limitations. They depend to a great extent on the population size, since the variance of the standardized rates is inversely proportional to the expected values; thus, areas with little population will present estimators with great variability. In this sense, the extreme standardized rates and, therefore, dominant in the apparent geographic pattern, are those estimated with the least precision in areas with few cases. (…)’

Some more relevant and recent literature need to mention in the introduction part as well as referencesection.

Thank you for the suggestion.

We have updated the Introduction section. On pages 3 and 4,

‘Our objective in this paper is to introduce an online interactive application that helps in real-time “Monitoring COVID-19 Pandemic in Catalonia, Spain”. It is worthy to mention that, since the outbreak of the pandemic, real-time interactive web applications have gained attention especially in the domain of monitoring and modeling dynamics of COVID-19. Throughout the globe several shiny dashboards are currently deployed and managed by individuals as well as government health organizations. Mukhtar et al. in their recent study have explored and analyzed a series of novel and interesting web-based applications that have been specifically developed during the pandemic that can act as a tool for the health professional’s community to help in advancing their analysis and research [5]. For example, Fernandez-Lozano et al. create an interactive dashboard to visualize all data related with the pandemic (cases, hospitalizations, and deaths) and its temporal evolution [6]. However, their tool does not display neither estimates nor other variables (environmental, vaccine status, etc.). The app created by Galván-Tejada et al. displays greater interactivity and completeness than the former but falls short of reporting updated vaccination data as well as other variables, even though they provide valuable demographic information on top of a very thorough analysis of the pandemic in Mexico [7]. Another useful online web application for updated country-specific analysis and visualization is “COVID19-World” by Tebé et al. used for basic epidemiological surveillance covering time trends and projections, population fatality rate, case fatality rate, and basic reproduction number [8]. Wissel et al. developed "COVID-19 Watcher," a similar web resource that displays COVID-19 data from every county and 188 metropolitan areas in the United States [9]. It provides the rankings of the worst-affected areas along with auto-generating plots depicting temporal changes in testing capacity, cases, and deaths. It is important to mention about “covid19.Explorer” R package and web application by Revell that has been designed to explore and analyze United States COVID-19 infection, death, and relative risk for different age groups with emphasis on geographic progress of the pandemic and effectiveness of lockdowns [10]. Similar interactive Python based analytical tool to compare data and monitor trends across geographical areas related to the COVID-19 pandemic across counties in the United States and worldwide is developed by Zohner et al. [11]. “Mortality Tracker” is another interesting in-browser application developed by Almeida et al. mainly focused on the visualization of public time series of COVID-19 mortality in the United States [12]. It was developed in response to requests by epidemiologists to access the effect of COVID-19 on other causes of death by comparing 2020 real time values with observations from the same week in the previous 5 years thus facilitate modeling the interdependence between its causes. Literature shows similar web application named “COVID19-Tracker” for Spain developed by Tobías et al. [13]. It produces daily updated data visualization and analysis of the COVID-19 diagnosed cases, and mortality in Spain. It also explores several analyses to estimate the case fatality rate, assessing the impact of lockdown measures on incident data patterns, estimating infection time and the fundamental reproduction number, and analyzing the mortality excess. An attempt of real-time statistical analysis in a user-friendly dashboard for researchers as well as general public is made by Salehi et al. [14]. It includes two mathematical methods (pandemic logistic and Gompertz growth models) to predict the dynamics of COVID-19, as well as the Moran's index metric, which provides a geographical perspective via heat maps and can help in the identification of latent reactions and behavioral patterns. Literature shows similar applications being implemented and maintained by researchers from various domains and different countries [15-18].

All these apps for respective regions systematically produce daily updated COVID-19 data visualizations and analysis. But a robust app with combination of relevant socioeconomic and environmental risk factors and their interrelation with the dynamics of the pandemic has been less explored. Thus, this complete project represents eight interactive dashboards which collectively enable ‘Monitoring the Pandemic in Catalonia (PandemonCAT)’ and explore environmental and socio-economic factors in the spatio-temporal evolution of the pandemic. 

It is a multicenter project, led by the Research Group on Statistics, Econometrics and Health (GRECS) of the University of Girona, Spain, in which the Andalusian School of Public Health (EASP) (Granada, Spain) and the University of Granada also participate. The aim of the PandemonCAT project is to provide a web application that allows the monitoring of the COVID-19 pandemic in Catalonia. Its results include, in addition to vaccination, the results of diagnostic tests, transmission (reproductive number), hospitalization, ICU admissions and the number of deaths. It also provides a visualization of those variables that could influence the spatio-temporal dynamics of the pandemic. Thus, in an interactive interface, the environmental variables (air pollutants and meteorological variables), the socio-economic variables, the points of interest where there may be a greater risk and the data of public mobility are shown.

The principal aim of the project is (…)’

In addition, we have added some relevant current references

15.- Berry, I.; Soucy, J.-P.R.; Tuite, A.; Fisman, D.; COVID-19 Canada Open DataWorking Group. Open Access Epidemiologic Data and an Interactive Dashboard to Monitor the COVID-19 Outbreak in Canada. Can. Med. Assoc. J. 2020, 192, E420

16.- Dong, E.; Du, H.; Gardner, L. An interactive web-based dashboard to track COVID-19 in real time. Lancet Infect. Dis. 2020, 20, 533–534

17.- Florez, H.; Singh, S. Online dashboard and data analysis approach for assessing COVID-19 case and death data. F1000Research 2020, 9, 1–13

18.- Marques da Costa, N.; Mileu, N.; Alves, A. Dashboard COMPRIME_COMPRI_MOv: Multiscalar Spatio-Temporal Monitoring of the COVID-19 Pandemic in Portugal. Future Internet 2021, 13, 45. https://doi.org/10.3390/fi13020045

Reviewer 2 Report

1.    The language is lucid, the idea is novel.
2.    This is a ‘good’ manuscript, but it prioritizes the methodology of data handling with an excessive use of technical jargon. The focus of the manuscript is not consistent with the title.
3.    There is considerable repetition of information – particularly in the introduction and methodology sections. Though minor, this should be amended.
4.    Table 1 is better suited to the supplementary.
5.    I would have liked to go through the coding used for the overall process. However, the web repository linked in the text does not provide the same.
6.    Some information can be reduced – for example, lines 50-69.
7.    Lines 115-122 is more well suited to the discussion/conclusion section.
8.    There are a few citation inconsistencies – e.g., line 138.
9.    Line 335 sounds vague and should be removed/rephrased.
10.    Regarding the use of data smoothing – the methodologies have not been explained in detail, or the process flows have not been adequately elaborated with reference to the present context.
11.     The information presented at lines 683-738 are more suitable for the introduction section since this information serves to present the lacunae in existing methodologies.

Author Response

Reviewer #2

  1.    The language is lucid, the idea is novel.

We thank the reviewer for their insightful comments.

  1.    This is a ‘good’ manuscript, but it prioritizes the methodology of data handling with an excessive use of technical jargon. The focus of the manuscript is not consistent with the title.

We have updated the Introduction, Discussion and Conclusion sections to emphasize on the title and focus on the main objective of the paper. All updates are highlighted in grey.

Thank you for the comment and suggestion.

  1.    There is considerable repetition of information – particularly in the introduction and methodology sections. Though minor, this should be amended.
  2.    Table 1 is better suited to the supplementary.

Thank you for the suggestion.

We have moved the table in the Appendix section and referred in the Introduction section. On page 4,

‘(…) Table 1 in Appendix reports information about individual dashboards of the application along with respective components and brief description.‘

  1.    I would have liked to go through the coding used for the overall process. However, the web repository linked in the text does not provide the same.

We will keep PandemonCAT code available in our www.researchprojects.es shortly. The code has designated blocks to download official data automatically to be used in the application. But following reviewer suggestion, in future we will make the entire pre-processed data also available in our website and GitHub as open data source. The datasets used in designing the application is already available in different heterogeneous official open data portals. But those data are raw and need extensive cleaning and pre-processing. We believe the compiled robust dataset can be useful for future researchers in this domain.

  1.    Some information can be reduced – for example, lines 50-69.

Following reviewer suggestion, we have updated and made it brief and precise.

On page 2

‘ Catalonia is basically an urban region. Sixty percent of the population resides in 23 cities with more than 50,000 inhabitants (comprising 6.62% of the total Catalan territory) and 52% in 14 cities with more than 100,000 inhabitants (comprising 4.64% of the territory) [3]. Among these include the second-largest city in Spain, Barcelona, and 36 adjacent municipalities making up the Barcelona Metropolitan Area. i.e.  41.75% of the population of Catalonia (representing only 1.97% of the territory). All have a high population density and exhibit different levels of urban air pollution whose primary source of emission is from road traffic. 

The administrative aggregation levels in Catalonia are the following, from highest to lowest: Autonomous Community (Catalonia), ‘Comarcas’ (counties-like regions), regions from now on (42 regions) and Municipalities (947 municipalities). Regions are widely used at an administrative level; they are even more relevant in the COVID-19 era with many restrictions affecting only certain regions. In addition, the health aggregation levels are the following, from highest to lowest: Autonomous Community, Health Region (9 health regions), Health sectors (20 health sectors), Health Basic Area (ABS, for its acronym in Catalan language) (343 ABSs).’

  1.    Lines 115-122 is more well suited to the discussion/conclusion section.

We have shifted these sentences in the Discussion section. On page 22,

‘(…) that could have some influence on the spatiotemporal dynamics of the pandemic. The app explores variables such as meteorological and air pollution variables, population by age group, unemployment rate, income per capita, and others in a geospatial interface. It also provides an interactive interface to visualize public mobility before, during, and after the lockdown phases in the community. The visualization of these variables could have some influence on the spatiotemporal dynamics of the pandemic.

On the other hand, linking the pandemic severity with environmental factors such (…)’

  1.    There are a few citation inconsistencies – e.g., line 138.

Thank you for pointing out this issue. We have updated it. On page 5,

‘(…) and plots internally. The key R packages used in the tool implementation include dplyr [20] and tidyverse [21] for data (…)’

  1.    Line 335 sounds vague and should be removed/rephrased.

We have removed the sentence.

  1.    Regarding the use of data smoothing – the methodologies have not been explained in detail, or the process flows have not been adequately elaborated with reference to the present context.

We are sorry to disagree with the reviewer. In section 2.4 we have tried to summarize the smoothing of rates. We believe that we have achieved a trade-off between extension (this is not an objective of the paper) and comprehension. However, we provide several references for the interested reader.

  1.     The information presented at lines 683-738 are more suitable for the introduction section since this information serves to present the lacunae in existing methodologies.

Thank you for the suggestion. We have moved lines (683-738) to the introduction section and have updated both introduction and discussion sections accordingly.

Reviewer 3 Report

The analyzed topic is very interesting and up to date in many contexts, e.g.: social, demographic, economic, administrative, spatial.

 The correct terminology was used. The language of the article is correct, adequate. 

The title is adequate to the research problem being undertaken. The article has been correctly divided into relevant sections, and their content coincides with their titles.

Most of the literature items presented in the reference list are current. They are all related to the topic presented.

Footnotes and bibliography are in my opinion correctly formulated.

The conducted research provides grounds for interesting conclusions. In my opinion, the conclusion part should be extended.

I do not fully understand the placement of table 1 in the paper. I think it is to too early. It should be in the following part. In my opinion, there is no more detailed information about individual applications. The table is very general. Aesthetically, table 1 is not well compiled. It uses a different font than the text, lines are left blank, etc. In terms of content, it is an interesting contribution to the topic of the paper.

The solution presented in the paper has a very application character. However, I do not know how it is rated by users and whether it was used and to what extent.

The interesting question is how will the application be used in the future and what data (and where from) will it be available in it?  

Author Response

Reviewer #3

The analyzed topic is very interesting and up to date in many contexts, e.g.: social, demographic, economic, administrative, spatial.

The correct terminology was used. The language of the article is correct, adequate. 

The title is adequate to the research problem being undertaken. The article has been correctly divided into relevant sections, and their content coincides with their titles.

 Most of the literature items presented in the reference list are current. They are all related to the topic presented.

Footnotes and bibliography are in my opinion correctly formulated.

The conducted research provides grounds for interesting conclusions. In my opinion, the conclusion part should be extended.

We thank the reviewer for their insightful comments.

I do not fully understand the placement of table 1 in the paper. I think it is to too early. It should be in the following part. In my opinion, there is no more detailed information about individual applications. The table is very general. Aesthetically, table 1 is not well compiled. It uses a different font than the text, lines are left blank, etc. In terms of content, it is an interesting contribution to the topic of the paper.

Thank you for the suggestion. We have moved the table in the Appendix section and referred in the Introduction section. We have also reformatted the entire table. On page 4,

‘(…) Table 1 in Appendix reports information about individual dashboards of the application along with respective components and brief description.‘

The solution presented in the paper has a very application character. However, I do not know how it is rated by users and whether it was used and to what extent.

Thank you for the comment.

We have hosted the application in “R Shiny” platform. In the R Shiny portal, we can monitor number of visits for the combined application as well as for individual component applications. The periodic report in the portal shows substantial number of visits.

Your suggestion provides us a new concept for the future to add a “Feedback” or, “Comments” section with existing About Us tab.

The interesting question is how will the application be used in the future and what data (and where from) will it be available in it?  

This is an interesting question.

We realised that, to prepare and give an effective public health response in the context to the pandemic, it is necessary to provide concrete, reliable, early and timely results for different affected populations. This is the principal motivation of our project. Moreover, if we present the official records in an interactive and open-access platform then the reach of the information will be widespread.

We had hosted the application during the second peak phase of the pandemic. During those phases and even now PandemonCAT application is being used as an open tool for consultation by the public of Catalonia and Spain in general. This will improve the understanding of the spread and the impact of COVID-19 in the community.

Moreover, the results of PandemonCAT and the robust and interrelated dataset used in the study can act as important source of baseline for future research works:

  • to explore the magnitude, spatiotemporal patterns and evolution of the impact and prevalence of COVID-19 disease in general and vulnerable population.
  • to analyse the impact of vaccination in a spatiotemporal context.
  • to understand the socioeconomic and environmental risk factors and their interrelation with the dynamics of the pandemic.
  • to explore the population dynamics (public mobility) in the spread in space and time of the infection and its incidence in mortality.
  • to investigate the association between air pollution and COVID-19.

We are keeping PandemonCAT code available in our www.researchprojects.es

The code has designated blocks to download official data automatically to be used in the application. But following reviewer suggestion, in future we will make the entire pre-processed data also available in our website and GitHub as open data source. The datasets used in designing the application is already available in different heterogeneous official open data portals. But those data are raw and need extensive cleaning and pre-processing. We believe the compiled robust dataset can be useful for future researchers in this domain.
